# Extreme Flooding Events in Coastal Lagoons: Seawater Parameters and Rainfall over A Six-Year Period in the Mar Menor (SE Spain)

Mariana Machado Toffolo [1,2,†], Federica Grilli [3,†], Catia Prandi [4,5], Stefano Goffredo [1,2] and Mauro Marini [2,3,*]

1 Marine Science Group, Department of Biological, Geological, and Environmental Sciences, University of Bologna, 40126 Bologna, Italy
2 Fano Marine Center, The Inter-Institute Center for Research on Marine Biodiversity, Resources and Biotechnologies, 61032 Fano, Italy
3 National Research Council, Institute for Biological Resources and Marine Biotechnologies (CNR, IRBIM), Largo Fiera della Pesca 2, 60125 Ancona, Italy
4 Department of Computer Science and Engineering, University of Bologna, 40126 Bologna, Italy
5 Polo Científico e Tecnológico da Madeira, (ITI/LARSyS), Caminho da Penteada, 9020-105 Funchal, Portugal
* Correspondence: mauro.marini@cnr.it
† These authors contributed equally to this work.

**Abstract:** Climate change is one of the main problems currently strongly conditioning ecosystems all over the world. Coastal lagoons are amongst the most vulnerable habitats, and they are undergoing extensive human impact due to their high production rates and the close proximity of urban and agricultural centers. The Mar Menor, the largest saltwater lagoon in Europe, is an example of a highly impacted ecosystem. In December 2016 and September 2019, climate change-induced DANA (upper-level isolated atmospheric depression) flooding events took place there, temporarily altering the lagoon oceanographic properties. Data gathered throughout the lagoon (11 stations inside and 1 outside the lagoon) from 2016 to 2021 were analyzed in order to assess the variability of seawater parameters: salinity, density, chlorophyll-*a*, turbidity, and dissolved oxygen, due to DANA events. Results showed a change in seawater parameters that were reestablished at different rates, 4 and 10 months in 2016 and 2019, respectively, following a description of the environmental conditions and effects that have been reported after extreme rainfall in the lagoon. The amount of rainfall correlated with changes in the analyzed seawater parameters, such as an increase in turbidity and chlorophyll-*a* values. Furthermore, turbidity correlated with chlorophyll-*a* and oxygen saturation, while density correlated with salinity. Such extreme weather events are worsened by climate change, growing more frequent and between shorter intervals in time. In order to decelerate ecosystem decline, comprehensive management plans are needed to address the various factors that might add to anthropic impacts in natural environments.

**Keywords:** DANA; temporal evolution; anthropization; torrential rain; Mediterranean Sea; coastal development

## 1. Introduction

The impact of human activities on climate change is un-neglectable. The current unwavering increase in global temperatures is leading to unprecedented changes, which could result in long-lasting, irreversible implications for ecosystems all over the world [1]. To date, 2019 was the second warmest year on record, with temperatures 1.15 °C higher than preindustrial values and more than double the average increase per decade (0.18 °C compared to 0.07 °C expected) [2]. These concerning values can promote substantial changes in natural parameters such as rainfall and sea level rise, leading to hurricanes, droughts, storms, wildfires, floods, and heatwaves, which in turn affect economic sectors

all around the globe [3,4], especially for populations and the ecosystems in regions subject to water stress [5].

The Mediterranean Sea is particularly vulnerable to these changes, as it is considered a climate change hotspot [1], subject not only to great rainfall variability throughout the region [6], with ever decreasing rainfall rates and an increase in evaporation with higher temperatures [7], but also to the risk of suffering extreme weather events (droughts, floods) [8], which contribute to the runoff and sedimentation of organic pollutants [9].

Of all the ecosystems impacted by climate change, coastal lagoons take special focus due to their rather fragile nature. Due to their transitional status (in between land and coastal waters), they are usually shallow and subject to extreme physical–chemical gradients, which in turn contribute to high levels of productivity, and they are consequently of major interest from an economic point of view [10,11]. The exploitation of coastal lagoons for various human activities, such as fishing, aquaculture, tourism, and sports, drainage basin use and agriculture put further stress into an already strained environment, subject to hydrodynamic, nutrient, and physical–chemical alterations that contribute to a decrease in environmental quality and natural resources [10,12].

One of the examples within the Mediterranean Sea is the Mar Menor, a hypersaline coastal lagoon (38.1–51) [13], one of the largest in the Mediterranean, with 135 km$^2$ in extension and a mean depth of 3.6 m [14]. The lagoon is situated in the Murcia region, in southeast Spain, at the end of a watershed bordered by a wide agricultural plain of approximately 1440 km$^2$ [15]. The lagoon is connected to the Mediterranean Sea through three shallow channels (Encañizadas del Ventorillo y La Torre, El Estacio, and Marchamalo) [15,16]. The mean temperature ranges from 30 °C during the summer to 11.2 °C in winter [13]. Annual rainfall is less than 300 mm year$^{-1}$, with evapotranspiration rates close to 900 mm year$^{-1}$ (hydrodynamic deficit of 600 mm year$^{-1}$) [13,15,17], and total water exchange every 318 days [18].

The high ecological importance and extensive impact contributed to official recognition as a susceptible area and the outset of several studies to mitigate the impacts and conserve the lagoon environment [19]. This key ecosystem has suffered further impact from a natural phenomenon that has been increasingly more severe over recent years, called DANA (Isolated Depression in High Levels). This phenomenon is characterized by masses of cold air that encounter the warmer Mediterranean air and produce heavy storms and intense rainfall [20]. It has been influenced by climate change [21], with changes in seasonality and water volume.

The goal of the present study is to assess the evolution of these effects of extreme flooding events induced by climate change in the Mar Menor.

## 2. Materials and Methods

From August 2016 to October 2021, the IMIDA (Murcian Institute of Agricultural Research & Development) field team surveyed 12 different points covering all areas of the Mar Menor (Figure 1). Utilized survey parameters were turbidity, chlorophyll-*a*, oxygen, and salinity. Values were obtained with a multiparameter profiler (SBE 19plus, Sea-Bird Electronics, WA, USA). Furthermore, rainfall data were obtained from the Sistema de Información Agrario de Murcia (SIAM) database (available at: http://siam.imida.es/apex/f?p=101:46:7220879812294039; Accessed on 12 July 2022).

Of all 12 sampling stations, 5 were selected based on their characteristics: E3, deeper situated in the "middle of the lagoon"; E4, which was closer to the ocean and far from the river flow, while being inside the lagoon; E5, which was outside the lagoon area, but in front of the lagoon; and finally, E7 and E8, which were closer to the drainage basin.

Regarding rainfall data, 7 stations were taken into consideration due to their location around rivers that flow into the lagoon (Figure 1). Monthly precipitation values for each of the stations, reported in the SIAM database, were used to calculate mean monthly rainfall.

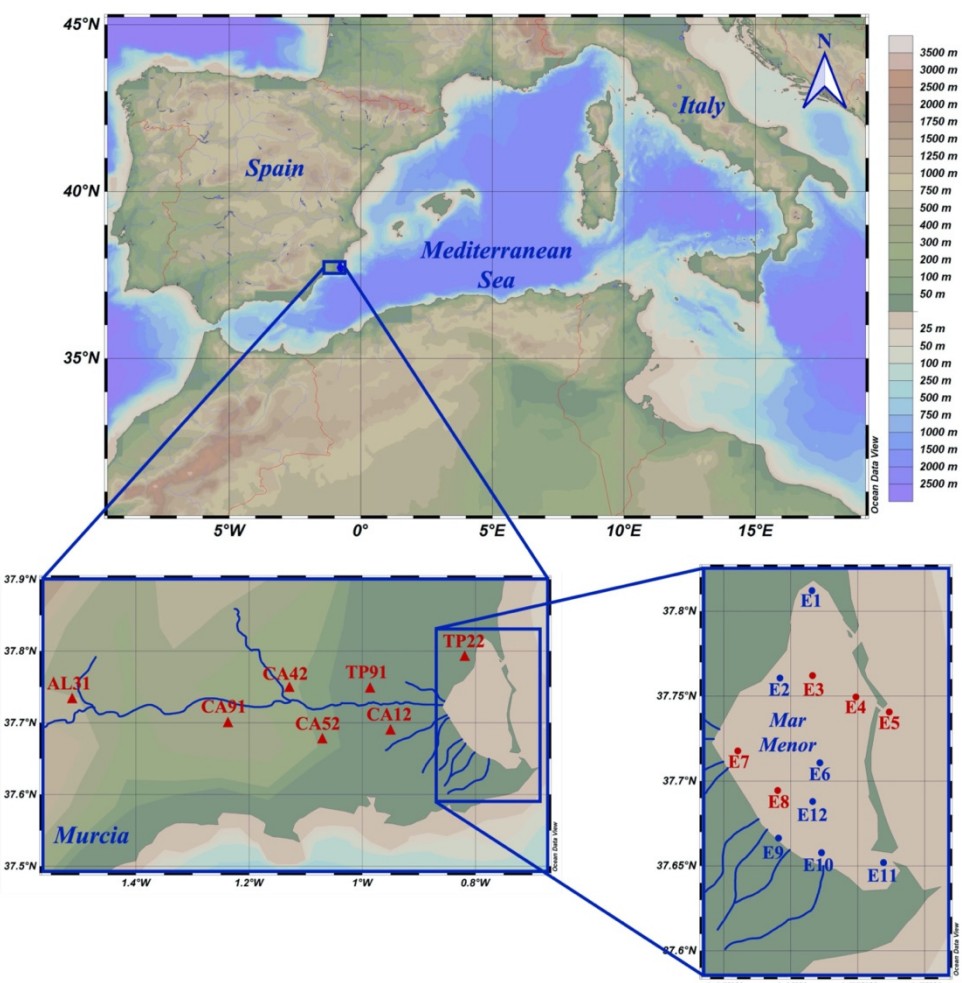

**Figure 1.** Location of the Mar Menor rainfall data stations (AL31: Lebor, Totana; CA12: La Palma, Cartagena; CA42: Balsapintada, Fuente Alamo; CA52: La Aljorra, Cartagena; CA91: Campillo Abajo, Fuente Alamo; TP22: Santiago de la Ribera, San Javier; TP91: Torre Pacheco, Torre Pacheco) and sampling stations within the lagoon territory. Sampling stations colored in red are the ones used for parameter analysis, based on their location (E3, E4, E5, E7, E8).

Seawater parameter analysis was carried out using the Ocean Data View (ODV) [22] software to create a timeline of the hydrological properties of the Mar Menor, comparing the parameters collected from the lagoon to rainfall values to identify the consequences of the two DANA events in December 2016 and September 2019. Dissolved oxygen (initially in mL L$^{-1}$, later transformed in % saturation) and potential density (obtained from temperature, depth, and salinity) parameters were calculated using the ODV software, which did not compute salinity values over 45. Higher salinity values were hence discarded to allow for parameter extrapolation.

Pearson's correlation coefficient analysis was performed using IBM SPSS Statistics v. 26 to test for relationships between rainfall and the analyzed parameters.

## 3. Results

The rates of monthly average rainfall in all seven data stations from August 2016 to October 2021 are shown in Figure 2a. Rainfall varies according to season, with four major peaks (where precipitation exceeded 100 mm month$^{-1}$) during the six-year period. These peaks in precipitation correspond to reported DANA events that took place during the periods 17–18 December 2016 and 12–13 September 2019, which reported 191 and 179 mm of rainfall, respectively. Further heavy rainfall events were observed, as seen in November 2018 (111 mm) and March 2020 (145 mm). After March 2020, there were no heavy rainfall

events observed. The sum of rainfall from September to May between 2016 and 2021, showing that 2019 saw the highest and extended value of precipitation (1366.82 mm) in comparison to the other years, can be observed in Figure 2b.

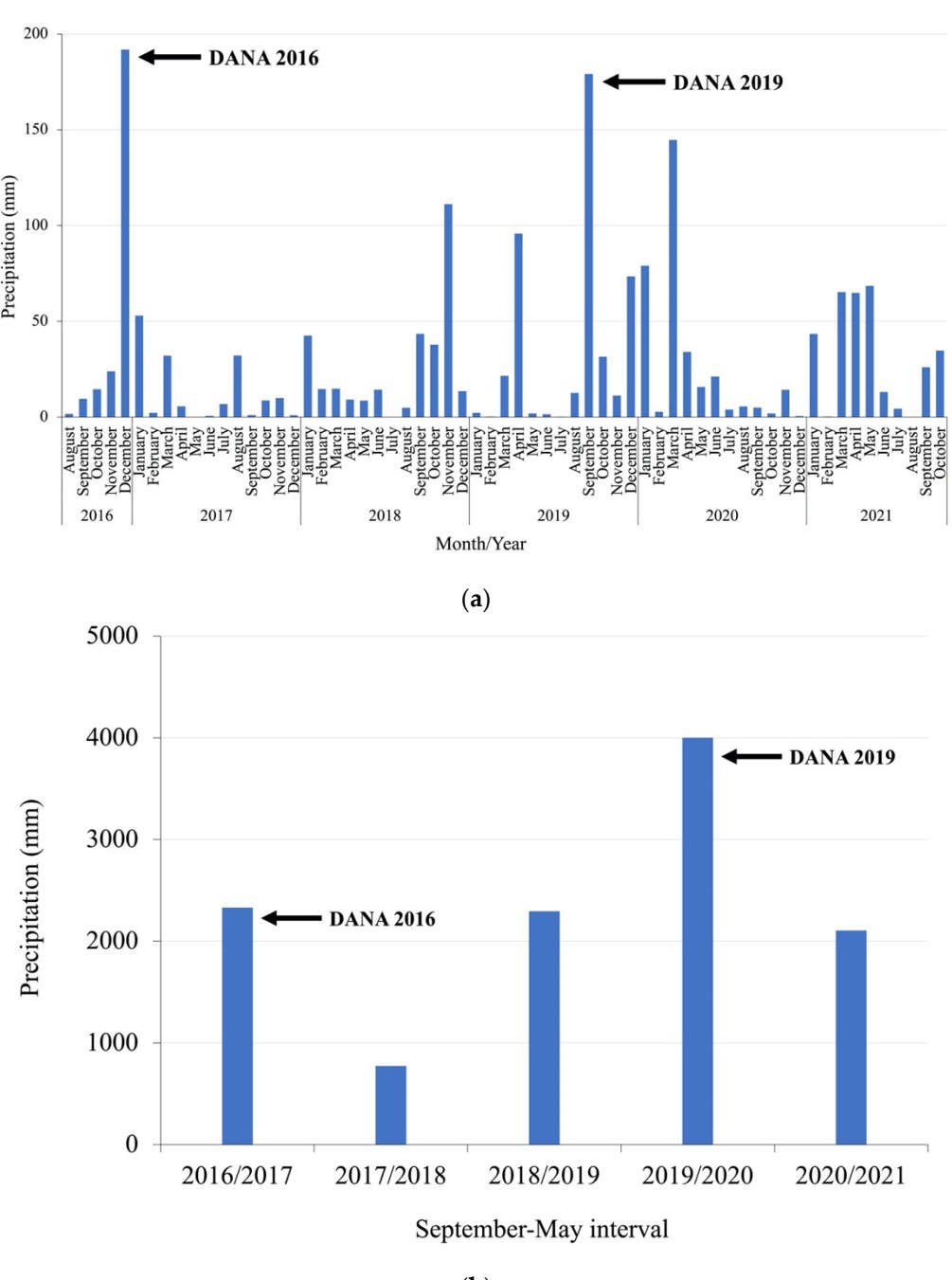

(**a**)

(**b**)

**Figure 2.** (**a**) Mean monthly rainfall for the 7 stations around the Mar Menor area from 2016 to 2021; (**b**) Precipitation values from September to May between 2016 and 2021.

The description of the temporal changes in salinity, density, dissolved oxygen saturation, chlorophyll-*a*, and turbidity focused on five stations inside the lagoon (stations E3, E4, E7, and E8) and one on the open sea (station E5). Inside the lagoon, salinity values (Figure 3), ranging from 38.5 to 47.2, are high compared to those in the open sea (37.0 to 43.5). There is an equal distribution of salinity, with high values (46.0 to 47.0) observed in 2016, followed by a swift decrease in 2017, related to the DANA in December 2016. There was a gradual increase in 2018, reaching higher salinity values during summer, followed

by an even greater decrease in 2019, corresponding to the DANA event that occurred in September 2019. These lower values were maintained until 2021, reinforced by another high precipitation event in March 2020, and further in March–May 2021. In the open sea, a peculiar stratification was observed: the surface layer was characterized by low salinity values (approximately 37.5 at 0–2.5 m), while there was a deeper layer with higher salinity values (>40 at 2.5–6 m), with mean salinity values of 34.5–39.0, as reported in other studies on the Mediterranean Sea [23,24].

## Salinity

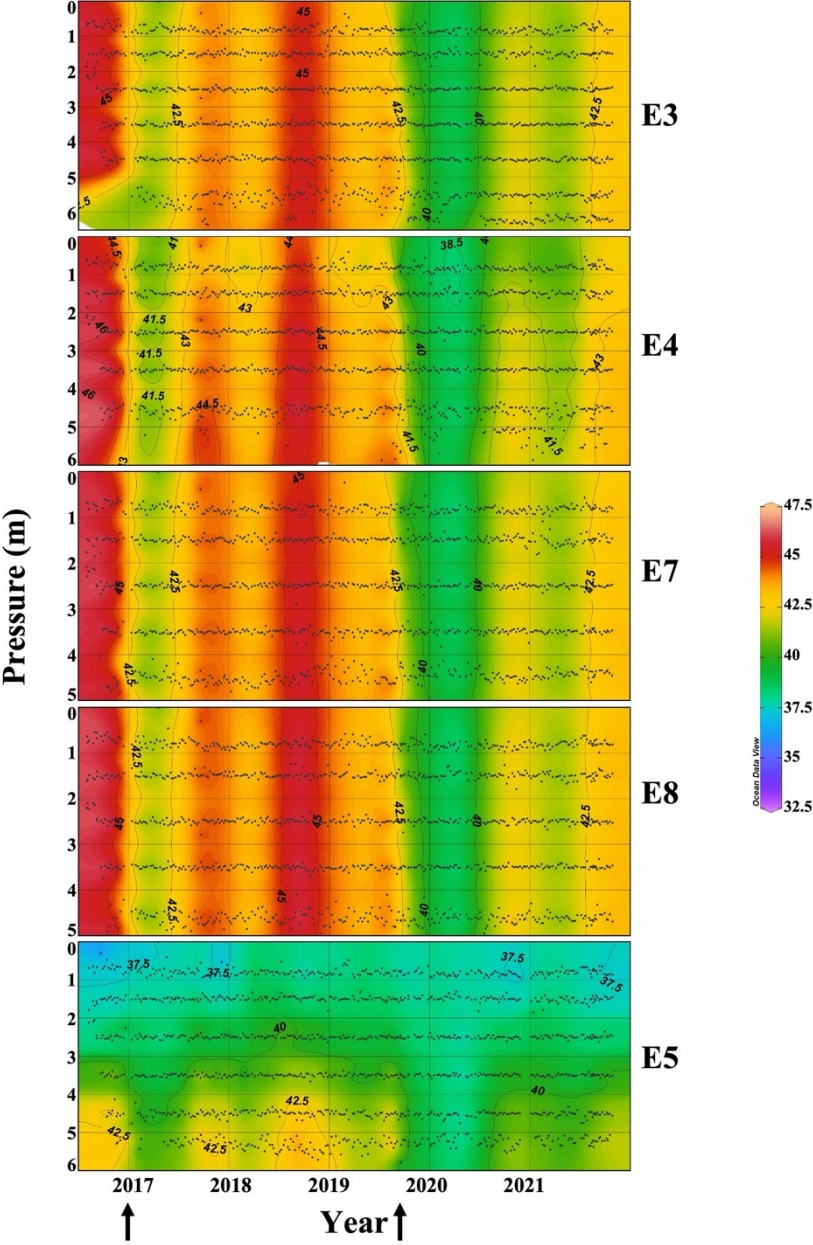

**Figure 3.** Temporal evolution of salinity at stations E3, E4, E7, E8, and E5 from August 2016 to October 2021. The vertical arrows indicate when the two DANA events occurred (December 2016 and September 2019, respectively).

The temporal evolution of potential density (Figure 4) for the stations inside the lagoon presented the same seasonal trend: higher density values (31–32 kg m$^{-3}$) in 2016,

with a decrease in early/mid-2017 (29–30 kg m$^{-3}$), an increase in late-2017/early-2018 (31.5–32.5 kg m$^{-3}$), a decrease in mid-2018 (30–31 kg m$^{-3}$), an increase in late-2018/early-2019 (31.5–32.5 kg m$^{-3}$), followed by a longer decrease in mid-2019 to mid-2020 (27–30.5 kg m$^{-3}$), another increase in late-2020/early-2021(28.5–31 kg m$^{-3}$), and finally decreasing in mid/late-2021 (28–20 kg m$^{-3}$). The temporal evolution of potential density in the open sea presented a different pattern in respect to the stations inside the lagoon. In fact, potential density increased with an increase in water depth (as for salinity), showing a surface layer (2.5 m depth) with values ranging from 25.5 kg m$^{-3}$ to 28 kg m$^{-3}$, and a deeper layer (2.5–6 m) with higher values, even reaching 30–31 kg m$^{-3}$.

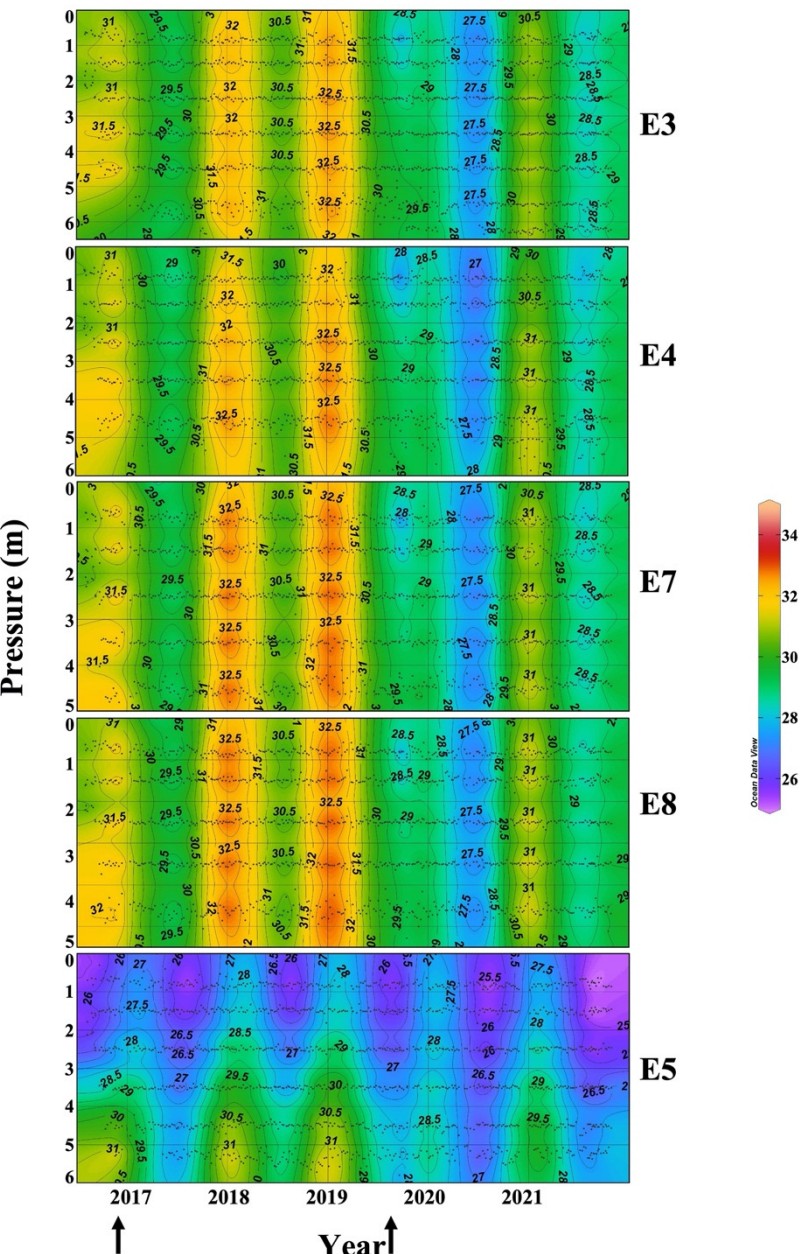

**Figure 4.** Temporal evolution of density at stations E3, E4, E7, E8, and E5 from August 2016 to October 2021. The vertical arrows indicate when the two DANA events occurred (December 2016 and September 2019, respectively).

Oxygen saturation (Figure 5) showed a rather homogeneous distribution throughout the sampling stations, with an overall supersaturated state (110–140%). In station E3, consistent fluctuations in saturation values were observed. During late-2019 and late-2021, there were lower saturation values at a depth of 5 m. Station E4 followed the same trend, with lower values present at a depth of 3 m in late-2017 and late-2021. E7 presented the highest saturation values (140–160%), particularly during mid-2018/mid-2019, and mid-2020/mid-2021. E8 saturation values were between 120% and 130% during 2016, showing higher saturation values in shallow depths (0–1 m) in 2017, 2019, and 2021, and at lower depths (3–4.5 m) in early-2019, 2020, and early-2021. Finally, in station E5, results showed fluctuations between seasons of the year from 2017 to 2019, while 2020 and 2021 presented higher saturation values throughout the year.

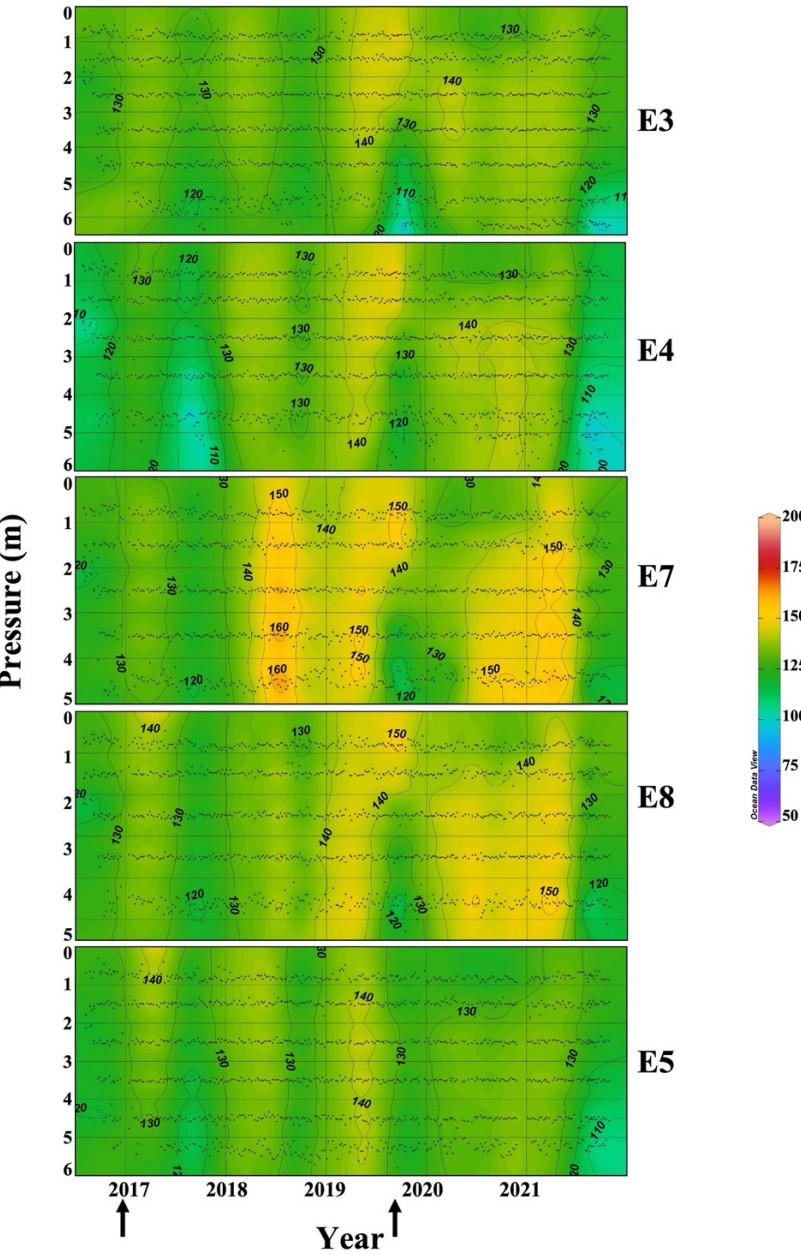

**Figure 5.** Temporal evolution of oxygen saturation at stations E3, E4, E7, E8, and E5 from August 2016 to October 2021. The vertical arrows indicate when the two DANA events occurred (December 2016 and September 2019, respectively).

A peak in chlorophyll-*a* (16–24 mg L$^{-1}$) was observed in all the stations inside the lagoon (Figure 6) during late-2016 and early-2017, followed by a decrease that remained constant (≤10 mg L$^{-1}$) throughout the following years. Station E5 showed constant lower values (≤10 mg L$^{-1}$) from 2016 to 2021, with higher values (10–14 mg L$^{-1}$) observed only during 2016, at 4–6 m depth.

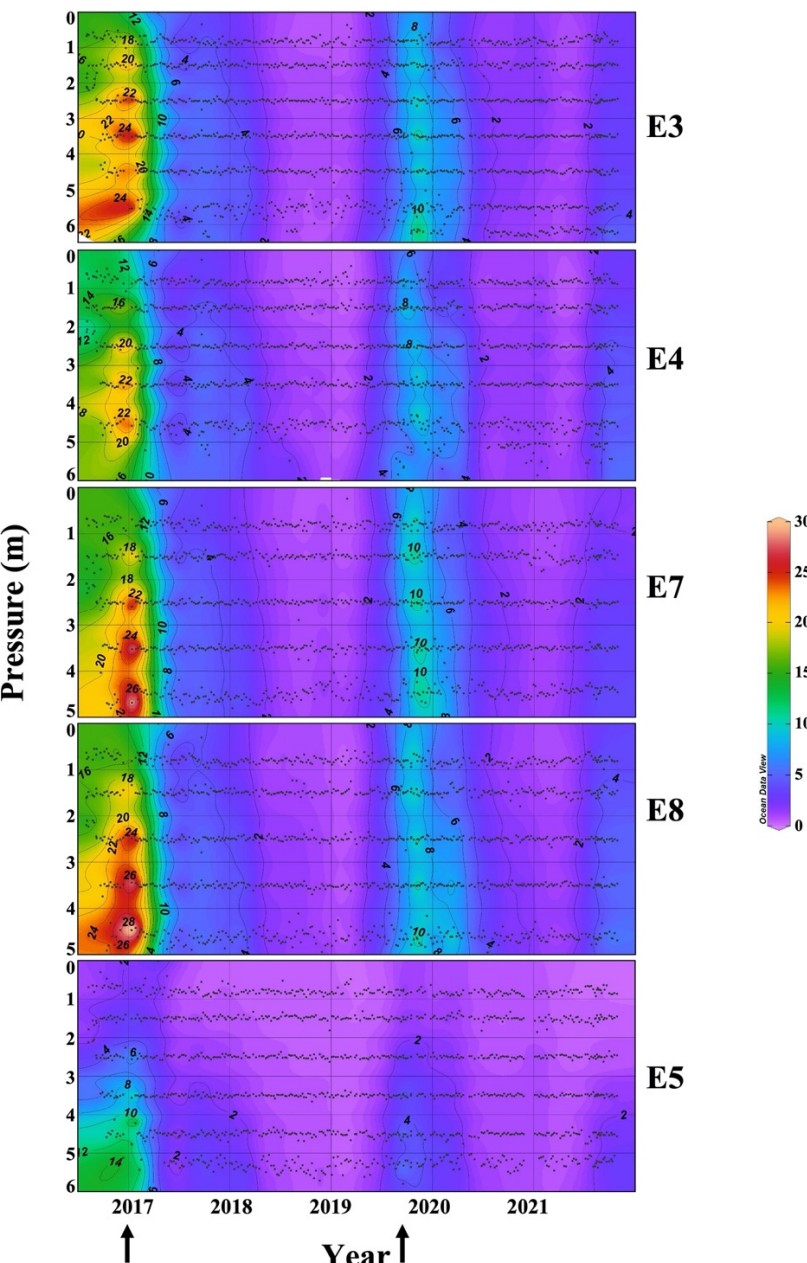

**Figure 6.** Temporal evolution of chlorophyll-*a* at stations E3, E4, E7, E8, and E5 from August 2016 to October 2021. The vertical arrows indicate when the two DANA events occurred (December 2016 and September 2019, respectively).

In stations E3 and E4 (Figure 7), higher turbidity values (4–5) were observed in 2016 and 2019/2020, in contrast with lower values during the other years (1–3). Station E7 showed higher turbidity values (4–10) during 2016, 2017, 2019/2020, and late-2021, with lower values (1–3) in 2018 and 2020/2021. Station E8 showed higher turbidity values (4–9) in 2016, late-2018, and 2019/2020, with lower values (1–3) in mid-2017, 2018, early-



2019. and late-2020/2021. In the open sea, higher turbidity values (4–6) were recorded in 2016 (down to 4 m depth) and in late-2019/early-2020. Lower turbidity values (1–3) were recorded from 2016 to early-2019, and late-2020/2021.

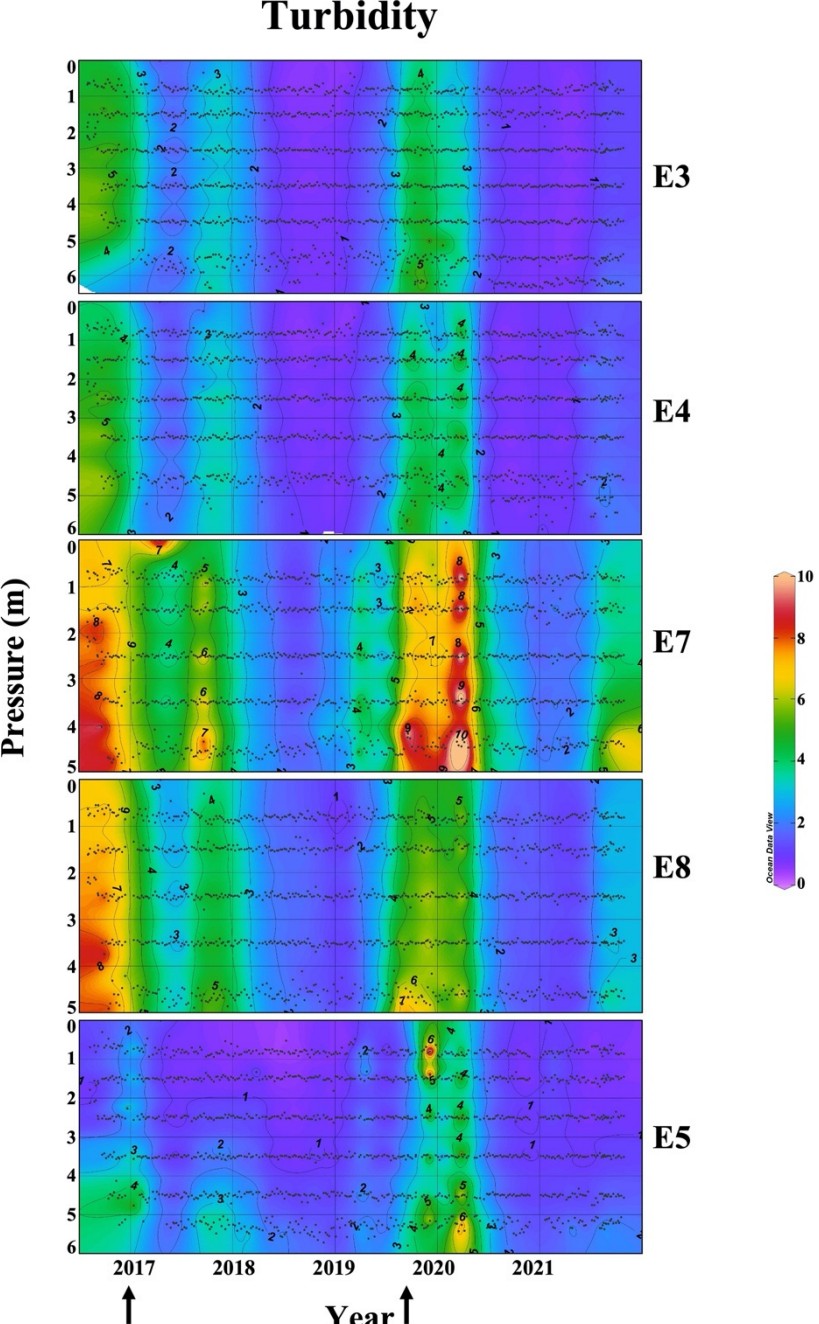

**Figure 7.** Temporal evolution of turbidity at stations E3, E4, E7, E8, and E5 from August 2016 to October 2021. The vertical arrows indicate when the two DANA events occurred (December 2016 and September 2019, respectively).

Table 1 reports Pearson's analysis between rainfall and seawater parameters and shows a significant correlation between rainfall and chlorophyll-*a* ($p < 0.05$), rainfall and turbidity ($p < 0.01$), turbidity and chlorophyll-*a* ($p < 0.001$), oxygen saturation and turbidity ($p < 0.05$), and density and salinity ($p < 0.001$, Table 1).

**Table 1.** Pearson's correlation efficient analysis. Significant values ($p < 0.05$) are displayed in bold. N = 63.

| | Rainfall | | Chlorophyll | | Turbidity | | Salinity | | Oxygen Saturation | |
|---|---|---|---|---|---|---|---|---|---|---|
| | $r$ | $r^2$ | $r$ | $r^2$ | $r$ | $r^2$ | $r$ | $r^2$ | $r$ | $r^2$ |
| Chlorophyll | **0.250** | **0.063** | | | | | | | | |
| Turbidity | **0.390** | **0.152** | **0.636** | **0.404** | | | | | | |
| Salinity | −0.120 | 0.014 | −0.020 | 0.000 | −0.136 | 0.018 | | | | |
| O2Sat | 0.100 | 0.010 | −0.178 | 0.032 | **−0.365** | **0.133** | −0.0253 | **0.001** | | |
| Density | 0.131 | 0.017 | 0.132 | 0.017 | −0.048 | 0.002 | **0.453** | **0.205** | 0.166 | 0.028 |

## 4. Discussion

The DANA events of 2016 and 2019 have had a significant impact in the environmental quality of the Mar Menor. All the analyzed parameters reported changes due to the extreme weather in all four analyzed sampling stations located inside the lagoon, with lesser consequences observed in the sampling station located outside the lagoon, in the open sea. Rainfall data reported the extreme inflow of water coming from land, exponentially increasing water exchange rates and carrying a variety of substances (organic pollutants, waste, sediments), causing the observed changes in the analyzed parameters throughout the lagoon (Figure 8). The input of nutrients carried from urban and agricultural areas into the lagoon is one of the main contributors to aquatic ecosystem contamination [9], promoting algal proliferation, causing shifts not only within the water column but also in the whole lagoon area. These shifts (especially at deeper depths) could in turn promote an oxygen crisis and subsequent eutrophication episode, severely compromising the equilibrium of the lagoon ecosystem [17].

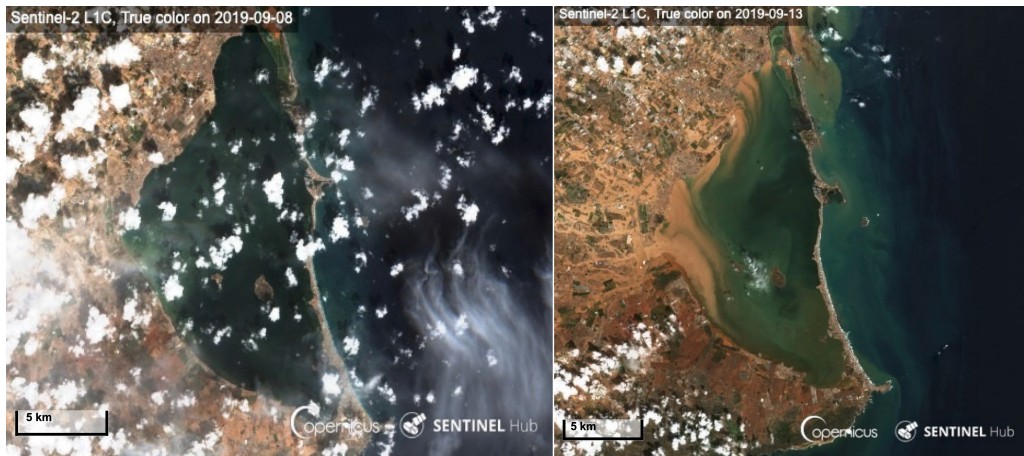

**Figure 8.** Satellite images of the Mar Menor before (**a**) and after (**b**) the DANA event of September 2019.

Density values are directly dependent on salinity, as reported in Table 1, which is influenced by the physical properties of the water body. Results showed that, in the open sea, there is a clear stratification in the water column, influenced by exchanges with the lagoon, where the water coming from the lagoon, with higher salinity (and consequently higher density) settles deeper in the column, while the water from the open sea, with lower salinity (and lower density) moves to shallower depths. The same was not observed within the lagoon, where instead the water column is more evenly distributed due to constant mixing.

Over the six-year period, the salinity values observed inside the lagoon were higher than in the open sea, corroborated with the precipitation regime of the area, with a deficit of 600 mm year$^{-1}$ between rainfall and evaporation [13,15,17], resulting in higher salinity values. These values were lowered during the reported DANA events in 2016 and 2019,

with an extreme inflow of freshwater that lowered salinity values for a period of 4 months in 2016/2017, and for a longer period of 10 months in 2019/2020. This could be explained by the higher volume of precipitation in 2019, followed by smaller but still intense rainfall events occurring in the following year that contributed to a decrease in salinity for a longer period of time. Consequently, density values followed the same trend, with lowest density values being observed following the DANA event in 2019.

The results showed a higher increase in chlorophyll-*a* values after the DANA event of December 2016, followed by a slightly subtler increase after the DANA in September 2019. The increase after 2019 is to be expected due to the inflow of runoff into the lagoon, which promoted growth within the algal community. However, the anomalous chlorophyll-*a* values observed in 2016/2017 can be explained by two factors: (i) the season is more favorable for the proliferation of algal biomass; (ii) a sudden change in water quality led to an eutrophication episode reported in January 2017 [17], where algal bloom reached its peak. Furthermore, there was an increase in turbidity values following the DANA events, particularly at stations E7 and E8, which were closest to the rivers that flow into the lagoon (Figure 9), presented higher turbidity values during that period in comparison with E3 and E4. Higher turbidity indicates sediments that can settle in the lagoon, carrying contaminants. The effect of this process is reflected in the contamination events that occur due to the rivers carrying pollutants through agricultural fields and urban areas [9].

Studies show that pollutants can be found in the water column, lagoon organisms, or sediments according to their physicochemical properties [25], and water bodies with higher salinity values have shown higher sorption and lower degradation rates of pollutants over time [26].

Regarding chlorophyll-*a* values, studies show that nitrogen is directly correlated with chlorophyll-*a* [27]. High nitrogen concentrations favor algal bloom, reflected in a high chlorophyll-*a* concentration. Hence, the observed peaks during 2017 and 2020 could be due to two factors: (i) The DANA events of 2016 and 2019, where the heavy rainfall carried nutrients (including nitrogen) from the agricultural fields in the surrounding Murcia region to the lagoon, correlated with high turbidity ($p < 0.01$, Table 1), favoring the proliferation of photosynthetic algae species, resulting in the increase of chlorophyll-*a* values [27,28], corroborated by the significant correlation indicated by Pearson's coefficient in Table 1; (ii) During the winter/early-spring season, chlorophyll-*a* values tend to rise to a peak [29], contributing to the high values observed on Figure 6. These results are in line with reported seasonal biomass peaks in transitional and coastal ecosystems [30], as well as eutrophication episodes in the Mar Menor [17].

Dissolved organic matter and algal biomass (measured through chlorophyll-*a* values), can contribute to the turbidity of the water column [31].

Although a decrease in oxygen means that the water is more stagnant, with less exchange of water between the surface and the bottom, and therefore the higher the movement of the water body, the higher the values of turbidity due to resuspension of organic matter in particulate sediment, algal biomass (chlorophyll-*a*), and oxygen saturation due to oxygenation of the water column [32], our results showed that turbidity is negatively correlated with oxygen saturation values. This could be due to higher microbiological activity through oxidation [33].

Boyer et al. [34] also reported that temperature has an influence on the values of dissolved oxygen, with lower saturation percentage with higher temperatures, and higher saturation percentage with lower temperatures.

According to the circulation patterns observed in the lagoon [13] (Figure 7), stations E3 and E4 have less water exchanges because they are only affected by winds and the circulation of the lagoon, while E7 and E8 are also conditioned by the rainfall and runoff inflow into the lagoon (Figure 8).

While it has been established that currents in the lagoon are motioned mainly by the wind (Figure 9) [13], the increase in turbidity could be caused by the DANA events, which brought about an extreme inflow of sediments and nutrients, with repercussions on algal

bloom, that reached the whole extension of the lagoon and also the boating canal that connects with the ocean, as shown in station E5.

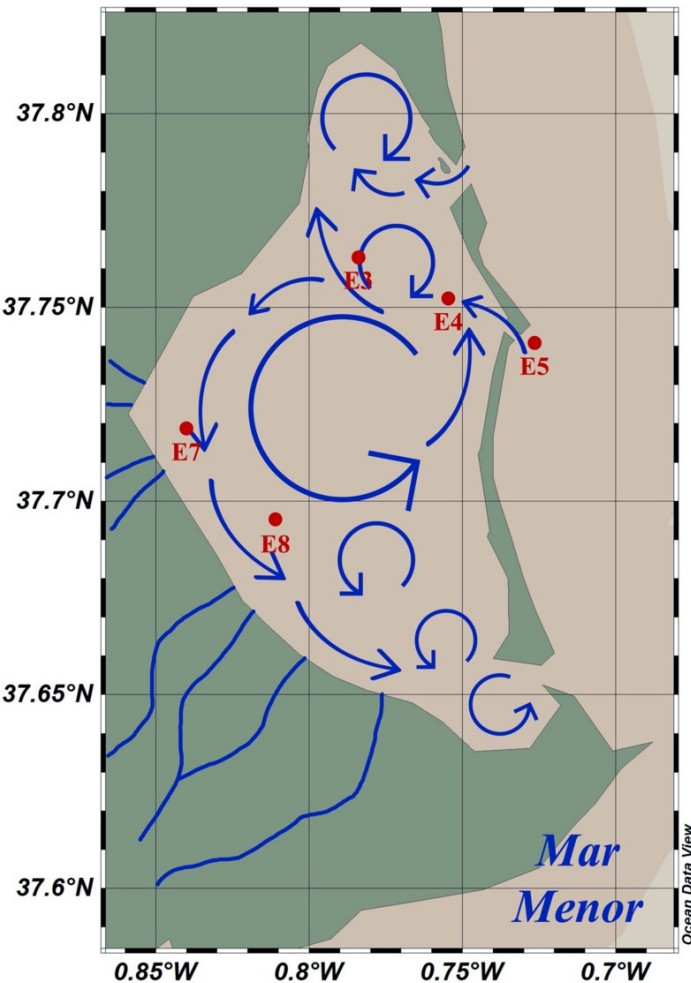

**Figure 9.** Current patterns, inflow watercourses, and canals connecting the Mar Menor to the Mediterranean Sea. Designed in accordance with Pérez-Ruzafa et al., 2005 [13], García-Oliva et al., 2018 [35], and Fernández-Alías et al., 2020 [36].

The Mar Menor is only one of innumerous similar ecosystems that are just as fragile and subject to extreme weather events. The Ria Formosa, a mesotidal lagoon in southern Portugal, is reported to suffer major disruptions in ecosystem balance due to extreme rainfall and oceanic upwelling [37]. The Rodrigo de Freitas Lagoon, in southwestern Brazil, described contrasting values in transparency, dissolved oxygen, and chlorophyll-*a*, as well as harmful bacterial presence following heavy rainfall rates [38]. Furthermore, extreme El Niño and La Niña events have been reported to influence the harvest period and exposure to pathogens in shellfish in the Mississippi Sound, USA [39]. Coastal lagoon management proposals have been carried out in the past, with the goal of maintaining the ecosystem services provided for humans. Nowadays, there is further need to expand such management to address climate change, which will not only affect ecosystem health but also bring consequences for the population in general [40].

## 5. Conclusions

Extreme weather events have long-lasting rippling impacts on the affected ecosystems. An unexpected increase in rainfall can have devastating effects, not only for urban areas with flooding and landslides but especially for natural ecosystems, which require a level of homeostasis that takes time and is very fragile to maintain. Climate change rates contribute

to such events and do not allow for ecosystem recovery, because the events are close in time and becoming ever more frequent. Because the physicochemical properties of a natural ecosystem are all interconnected, these circumstances create a cascade effect that can permanently compromise their functioning and be detrimental, not only for the ecosystem survival but also for the plethora of organisms that directly or indirectly depend on it, of which the human race is one.

Because coastal lagoons are highly productive albeit fragile ecosystems, the presented seawater parameter analysis in contrast with extreme weather events might serve as a pilot study that can be further applied and tailored to fit different environmental contexts around the world.

These analyses can assist in designing the mitigation actions needed to cope with climate change rates within damaged territories, as well as comprehensive management plans that take into consideration all aspects that might contribute to the decline of natural ecosystems due to pollution from agricultural and urban area runoff.

**Author Contributions:** Conceptualization, M.M.; methodology, F.G.; validation, M.M.; formal analysis, M.M.T.; investigation, M.M.T. and F.G.; data curation, M.M.T., F.G. and M.M.; writing—original draft preparation, M.M.T., F.G. and M.M.; writing—review and editing, M.M.T., F.G. and M.M.; visualization, F.G.; supervision, C.P. and S.G. All authors have read and agreed to the published version of the manuscript.

**Funding:** This research received no external funding.

**Institutional Review Board Statement:** The study did not require ethical approval.

**Informed Consent Statement:** Not applicable.

**Data Availability Statement:** All data used on this study can be found in both the *Sistema de Información Agrario de Murcia* (SIAM) database (available at: http://siam.imida.es/apex/f?p=101:46:7220879812294039, access on 1 September 2022) and in the Mar Menor information service website (Canal Mar Menor, available at: https://canalmarmenor.carm.es/, access on 1 September 2022), provided by *the Comunidad Autonoma de la Región de Murcia* (CARM).

**Acknowledgments:** This study represents partial fulfilment of the requirements for the doctoral thesis of M. Machado Toffolo, within the international Program "Innovative Technologies and Sustainable Use of Mediterranean Sea Fishery and Biological Resources" (FishMed-PhD; www.FishMed-PhD.org) at the University of Bologna, Italy. The authors acknowledge the *Comunidad Autonoma de la Región de Murcia* (CARM) for the data provided through the Mar Menor information service website (Canal Mar Menor).

**Conflicts of Interest:** The authors declare no conflict of interest.

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
