# Peer review of "Extreme Flooding Events in Coastal Lagoons: Seawater Parameters and Rainfall over A Six-Year Period in the Mar Menor (SE Spain)"

_jmse, doi:10.3390/jmse10101521_

Round 1

Reviewer 1 Report

The paper presents a good visualization of years of monitoring results of the environmental conditions in the study area. The data visualization is clear, and the results are well supported by the data presented. The methodology is sound, and the conclusions are justified. The only suggestion is to explain more the implications for environmental management, and especially how to respond to the impacts of climate-induced DANA phenomena.

Author Response

Response to Reviewer 1 

Point 1: The paper presents a good visualization of years of monitoring results of the environmental conditions in the study area. The data visualization is clear, and the results are well supported by the data presented. The methodology is sound, and the conclusions are justified. The only suggestion is to explain more the implications for environmental management, and especially how to respond to the impacts of climate-induced DANA phenomena.

 Response 1: We thank the reviewer for the suggestion. We have sought to address the reviewer’s comment in the discussion and conclusion sections.

Reviewer 2 Report

The manuscript by Machado et al. provides very interesting and useful data on the changes experienced by the coastal lagoon of Mar Menor after the impact of two DANAS in recent years. This theme is scarcely studied and, considering the high ecological value of this area and the intensification and increase in the frequency of torrential rains and high-energy marine events in the Mediterranean basin in the context of global change, this work is undoubtedly a significant contribution.   However, in my opinion, its structure and discussion is excessively simple . A statistical analysis could have been applied to the dataset to look for correlation between the measured parameters. In adittion, a more profuse discussion is necessary with comparison of similar coastal lagoons also affected by DANASn the Mediterranean and other parts of the world. The approach of a conceptual model of changes in the lagoon after the DANAs, and the of implications of these events  on ecosystems and human activity until the complete recovery of the initial conditions, is also necessary.    In my opinion the paper can be accepted with moderated revisions if the proposed changes and suggestions included in the revised manuscript (pdf attached) are carried out and the discussion is expanded and improved.  

Author Response

Response to Reviewer 2 

Point 1: The manuscript by Machado et al. provides very interesting and useful data on the changes experienced by the coastal lagoon of Mar Menor after the impact of two DANAS in recent years. This theme is scarcely studied and, considering the high ecological value of this area and the intensification and increase in the frequency of torrential rains and high-energy marine events in the Mediterranean basin in the context of global change, this work is undoubtedly a significant contribution. However, in my opinion, its structure and discussion is excessively simple . A statistical analysis could have been applied to the dataset to look for correlation between the measured parameters. In adittion, a more profuse discussion is necessary with comparison of similar coastal lagoons also affected by DANAS the Mediterranean and other parts of the world. The approach of a conceptual model of changes in the lagoon after the DANAs, and the of implications of these events on ecosystems and human activity until the complete recovery of the initial conditions, is also necessary. In my opinion the paper can be accepted with moderated revisions if the proposed changes and suggestions included in the revised manuscript (pdf attached) are carried out and the discussion is expanded and improved.

Response 1: We thank the reviewer for the comments and suggestions. We sought to complement the Discussion in the light of correlation measures and comparison with other similar ecosystems, as suggested by the reviewer, and also address the comments made in the attached pdf document.

Reviewer 3 Report

The manuscript titled "Extreme flooding events in coastal lagoons: oceanographic parameters and rainfall over a 6-year period in the Mar Menor (SE Spain)" aims to investigate more in general the effects of Climate change on coastal lagoons. In particular, the authors have developed the research in the study site of Mar Menor, analyzing the direct effects of upper-level isolated atmospheric depressions flooding events (DANA) to the oceanographic properties of the lagoon.

The abstract is well connected with the next sections of the paper, and some numerical results are shown, helping to read the graphs and stimulate the reader to find out the conclusions.   

Introduction (section 1) is exhaustive, and I have appreciated the bibliography, that is updated to the most recent publications, and the most relevant to the Mar Menor area.

The Materials and Methods section (2) is concise enough, but Figure 1 should be integrated with rivers and catchments basins, and eventually few isolines that would give an idea about the orography and the relative location of the stations. Of course, this is valid depending on data availability, but in case that the plots will be enriched, an appropriate scale bar should be added.

The attempt has been done by the authors in the text lines from 93 to 100, but the description sometime is dispersive, especially for the readers that are not confident with Spanish names and do not know the places mentioned.

Even to describe distances between stations and water bodies such as "far" or "close" does not give an objective observation. For those reasons I think that improving the Figure 1 will make this section clearer, and the few items to add can maybe suggest future development of the research.

Results (section 3) are well exposed. Interline between the caption of Figure 2 has to be adjusted.

Even the Discussion chapter (4) is well described. The references to the research’s pillars are well distributed, and supportive of the authors’ theories and findings.

Similarly, to Figure 1, in Figure 9 some details about river bodies and urbans/ anthropogenic potential sources of pollutants should be added. Integrating the Figure1 could also be useful to better read Figure 9.

Conclusions section (5) is brief, and well connected with discussion and abstract. They do not explicitly  refer to the case study, but brings the problem to a greater scale. 

In general, the manuscript is well described and structured. English is clear, well written and understandable. The topic is very interesting and indispensable during the present. I found the paper interesting even because of its applicability worldwide, that should be emphasized in the paper.

Author Response

Response to Reviewer 3 

Point 1: The manuscript titled "Extreme flooding events in coastal lagoons: oceanographic parameters and rainfall over a 6-year period in the Mar Menor (SE Spain)" aims to investigate more in general the effects of Climate change on coastal lagoons. In particular, the authors have developed the research in the study site of Mar Menor, analyzing the direct effects of upper-level isolated atmospheric depressions flooding events (DANA) to the oceanographic properties of the lagoon.

The abstract is well connected with the next sections of the paper, and some numerical results are shown, helping to read the graphs and stimulate the reader to find out the conclusions.

Introduction (section 1) is exhaustive, and I have appreciated the bibliography, that is updated to the most recent publications, and the most relevant to the Mar Menor area. The Materials and Methods section (2) is concise enough, but Figure 1 should be integrated with rivers and catchments basins, and eventually few isolines that would give an idea about the orography and the relative location of the stations. Of course, this is valid depending on data availability, but in case that the plots will be enriched, an appropriate scale bar should be added. The attempt has been done by the authors in the text lines from 93 to 100, but the description sometime is dispersive, especially for the readers that are not confident with Spanish names and do not know the places mentioned. Even to describe distances between stations and water bodies such as "far" or "close" does not give an objective observation. For those reasons I think that improving the Figure 1 will make this section clearer, and the few items to add can maybe suggest future development of the research.

Response 1: Agreed. We thank the reviewer for the suggestion. We have added relevant information concerning the catchment basin and inflow rivers within the Mar Menor area, and also regarding the orography of the region, in order to clarify the information on the survey stations’ location within the text in Figure 1. Regarding the scale bar for distance, we have considered longitude degrees, represented on the border of the figures, as a measure of distance.

Point 2: Results (section 3) are well exposed. Interline between the caption of Figure 2 has to be adjusted. Even the Discussion chapter (4) is well described. The references to the research’s pillars are well distributed, and supportive of the authors’ theories and findings.

 Response 2: We thank the reviewer for the comment. We have corrected the caption interline of Figure 2 as suggested.

 Point 3: Similarly, to Figure 1, in Figure 9 some details about river bodies and urbans/ anthropogenic potential sources of pollutants should be added. Integrating the Figure1 could also be useful to better read Figure 9.

 Response 3: Agreed. We have added relevant information concerning the catchment basin and inflow rivers within the Mar Menor area to Figure 9 as well.

Point 4: Conclusions section (5) is brief, and well connected with discussion and abstract. They do not explicitly refer to the case study, but brings the problem to a greater scale.

In general, the manuscript is well described and structured. English is clear, well written and understandable. The topic is very interesting and indispensable during the present. I found the paper interesting even because of its applicability worldwide, that should be emphasized in the paper.

 Response 4: We thank the reviewer for the suggestion. We sough to expand the discussion/conclusion sessions to address the study’s applicability, as suggested.

Round 2

Reviewer 2 Report

The most relevant changes in form and content have been carried out in the manuscript and, in its current form. Only a few small changes in Table 1 are necessary for final acceptance: to delete N (can be included in the table caption), put the significant r values in bold and remove “p” from the table. You need also to raise the value of "r" to the square (r2).

Regards

Author Response

Response to Reviewer 2 Comments

Point 1: The most relevant changes in form and content have been carried out in the manuscript and, in its current form. Only a few small changes in Table 1 are necessary for final acceptance: to delete N (can be included in the table caption), put the significant r values in bold and remove “p” from the table. You need also to raise the value of “r” to the square (r2).

 Response 1: We thank the reviewer for the comments. We have deleted N from Table 1, included it in the table caption, put significant r values in bold, removed “p” and added the value of “r2 to the table.